# Culturally Tailored Community Brain Health Education for Chinese Americans Aged 50 or Above: A Mixed-Methods Open Pilot Study

**DOI:** 10.3390/geriatrics10020058

**Published:** 2025-04-14

**Authors:** Kaipeng Wang, Fei Sun, Peiyuan Zhang, Carson M. De Fries, Xiaoyouxiang Li, Jie Zhu, My Ngoc To

**Affiliations:** 1Graduate School of Social Work, University of Denver, Denver, CO 80208, USAmyngoc.to@du.edu (M.N.T.); 2School of Social Work, Michigan State University, East Lansing, MI 48824, USA; sunfei@msu.edu; 3School of Social Work, University of Maryland, Baltimore, MD 21201, USA; peiyuanzhang@ssw.umaryland.edu; 4Morgridge College of Education, University of Denver, Denver, CO 80208, USA; xiaoyouxiang.li@du.edu; 5Nutrition and Foods Program, School of Family and Consumer Sciences, Texas State University, San Marcos, TX 78666, USA; j_z151@txstate.edu

**Keywords:** brain health, Chinese Americans, community health education, mixed methods

## Abstract

**Background:** Chinese Americans, the largest Asian American subgroup in the U.S., face linguistic, cultural, and socio-economic barriers to dementia prevention. To promote brain health in this population, a culturally tailored community approach is essential. This study evaluates a culturally tailored community brain health education program to enhance brain health knowledge and motivate lifestyle changes to prevent the risk of dementia among Chinese Americans aged 50 or older. **Methods:** The program was developed and evaluated in four phases. First, we assessed participants’ interests in brain health topics, availability, and preferred delivery modes. Next, experts on the identified topics developed educational content and outcome assessments. The third phase focused on implementing a six-session program covering general knowledge about Alzheimer’s disease and related dementias, diet, sleep, physical exercise, health checks, and mindfulness. Finally, we evaluated the program’s feasibility and effectiveness using pre–post surveys, feedback questionnaires, and focus groups. **Results:** Seventy-seven participants registered for the program, and sixty-nine (90%) attended at least four sessions. The quantitative results, based on paired *t*-tests, showed significant increases in brain health knowledge, sleep quality, and behavioral motivation for lifestyle changes, and a decrease in depressive symptoms, with two-tailed *p*-values lower than 0.05. The qualitative results further revealed promising feasibility and acceptability, as well as the perceived benefits of the program. **Conclusions:** The findings highlight the feasibility, acceptability, and potential effectiveness of a culturally tailored community education approach for promoting brain health and lifestyle changes. Sustained community outreach and education efforts among Chinese Americans are needed.

## 1. Introduction

### 1.1. Study Background

Dementia continues to present a global public health challenge. In 2020, the number of people living with dementia reached over 55 million, which is estimated to reach 139 million in 2050 [1]. The United States is not exempt from this issue, with an estimated 4.0% prevalence of diagnosed dementia among adults aged 65 and older, and 13.1% among those aged 85 and older [2]. While nationally representative data on dementia prevalence among Chinese Americans are lacking, a systematic review of non-nationally representative studies identified two reports estimating the prevalence of Alzheimer’s Disease and related dementias (ADRD) among Chinese Americans aged 65 or older at 13.3% and 16.7%, respectively [3]. More importantly, this separate systematic review and meta-analysis not only identified an average 10.9% prevalence rate of ADRD in older Asian Americans, primarily Chinese Americans, but also underscored the systematic underrepresentation of Asian Americans in ADRD research [3].

Research has shown that modifiable lifestyle factors such as diet, physical activity, chronic condition management, and isolation are significantly associated with dementia risk, even after accounting for genetic factors [4,5]. Brain health, referring to “a state of complete physical, mental, and social well-being through a full, balanced, continuous development and exercise of the brain”, is increasingly recognized as critical in reducing dementia risk [6]. Community education plays a key role in public health by promoting healthy lifestyles and prevention strategies [7]. Modifiable risk factors remain a focus because they stem from the complex interplay of individuals’ history, background, and unique lifestyles [7]. Addressing these factors through tailored community education programs offers flexibility and the ability to adapt over time based on new research [8]. Such programs empower individuals to make informed decisions, thereby reducing dementia risk and alleviating related societal and economic burdens [9]. Therefore, brain health education aimed at improving modifiable lifestyles is critical for dementia risk prevention. Several studies have shown that brain health education targeting dementia prevention for the general population has improved knowledge of dementia, self-efficacy for performing daily tasks, health motivation, and healthy lifestyle behaviors [10,11,12]. While none of these studies have centered older Chinese Americans, the benefits of such educational programs are likely transferrable, especially when the challenges, needs, and preferences of older Chinese Americans are incorporated into program development [13,14].

Older Chinese Americans, especially those with limited English proficiency, face unique barriers to accessing health promotion programs. Specifically, cultural and linguistic challenges hinder their participation and engagement in these educational programs [15]. For instance, older Chinese immigrants heavily rely on media outlets in their native language or dialect to receive day-to-day information, including educational information about brain health [16]. Unfortunately, health information is often unavailable in a Chinese language or dialect [17]. In addition to language barriers, a lack of cultural tailoring may further hinder the delivery and effectiveness of health-related educational content [18,19]. These barriers necessitate the development of culturally sensitive and linguistically appropriate educational interventions. A critical strategy for effective cultural tailoring involves stakeholder-centered and needs-based approaches [20,21], which were integral to the design of the present project.

Several studies have investigated the utilization of culturally tailored programs, including online videos and websites for brain health education and a brain health awareness campaign designed for Chinese Americans [22,23,24]. Although these studies address important gaps in promoting brain health knowledge and community engagement among older Chinese Americans, to our knowledge, no existing study has evaluated a structured and culturally tailored community-based brain health education program for this population. Such a gap is especially salient in geographic regions where health promotion resources are not easily accessible to older Asian Americans due to linguistic and cultural barriers.

### 1.2. Study Aims

The primary aim of this study is to examine the feasibility of a culturally tailored community brain health education program for Mandarin-speaking Chinese Americans aged 50 or older in the Denver Metro Area, where no such program has previously been offered. The secondary aim is to explore the program’s effectiveness in improving participants’ knowledge, behavioral motivation, short-term health outcomes, and dementia worry associated with brain health after program completion.

Our decision to include individuals aged 50 and above reflects a proactive, preventive approach to brain health. Targeting this age group allows us to introduce educational interventions before the typical age of increased dementia prevalence, potentially delaying or preventing onset. This approach aligns with current public health strategies recommended by the American Association of Retired Persons (AARP), which emphasize early education and lifestyle modifications as critical components in reducing dementia risk after 50 [25].

While we acknowledge the needs of Chinese Americans who do not speak Mandarin or live outside this area in the Greater Denver Metro Area, our focus on Mandarin-speaking Chinese Americans in the Greater Denver Metro Area is deliberate for several reasons. First, compared to English-speaking Chinese Americans, this demographic faces additional cultural and linguistic barriers to accessing conventional health education and services. Second, there is a lack of research on brain health education within this community, underscoring the need for targeted intervention. Third, the research team has established community connections in the Denver area, which primarily serve older Chinese Americans who speak Mandarin only.

## 2. Materials and Methods

### 2.1. Participants and Recruitment

To be eligible for this study, participants had to (1) be at least 50 years old, (2) self-identify as of Chinese ethnicity, (3) have the capacity to communicate in Mandarin Chinese in speaking and writing, and (4) reside in the Greater Denver Metro Area. Participants were recruited through community events, social media platforms, and word-of-mouth referrals. Interested participants met with a research team member to confirm their eligibility and consent to participate in the study. Figure 1 presents the study participant flow chart.

### 2.2. Design Overview and Study Procedure

This study was approved by the University of Denver Institutional Review Board (Project Number: 2048378-1, approval date: 11 May 2023). A convergent parallel mixed-methods approach was used to not only quantitatively assess the effectiveness of the program but also gain a deep understanding of how the program impacted participants’ knowledge and lifestyle related to brain health and identify areas for improvement. The quantitative component served two objectives: (1) assessing program feasibility and (2) exploring program effectiveness. A single group pre-test and post-test design was utilized to compare outcomes such as brain health knowledge, behavioral motivation for lifestyle changes related to brain health, and short-term health outcomes associated with brain health at baseline and two weeks after the program ends. Participants were asked to complete a 15–20 min pre-test survey at baseline and received a 20 USD gift card for completing the pre-survey. To assess program feasibility, attendance was recorded for each program session. Participants who completed at least four sessions were invited to complete a 15–20-min post-test survey two weeks after the program ended and received a 20 USD gift card for completing the post-survey. Surveys were distributed either in-person or online, depending on the participants’ preferred mode of educational delivery.

The qualitative component had three objectives: (1) providing additional insights into the program’s feasibility, (2) assessing its impact on participants’ knowledge and behavioral motivation for lifestyle changes related to brain health and dementia worry, and (3) identifying potential areas for improvement of the program. Qualitative data were collected through session feedback questionnaires completed at the end of each session and follow-up focus group interviews.

Immediately after each session, participants were invited to complete a short feedback questionnaire for the session, which contained three open-ended questions: (1) What did you learn from the workshops or lectures? (2) What are some strengths of the workshops or lectures? (3) What are some possible improvements for the workshops or lectures? The purpose of the questionnaire was two-fold: it provided an opportunity for participants to review and reflect on each session and for the research team to understand the participants’ experience of the session and the impact of each session’s educational content. During each session, participants who completed the survey were entered into a raffle in which 20 of them were drawn to receive a 10 USD gift card.

Two weeks after the program ended, participants who attended at least four out of six sessions were invited to participate in an in-person or online focus group interview to provide more detailed insights on their experiences and perceptions of the program’s feasibility. A convenience sampling approach, based on a first-come, first-served method, yielded 17 participants, with 10 joining the in-person group and seven attending the online group. Each interview was conducted by the principal investigator (PI) and a bilingual research assistant who has experience in facilitating focus groups. Semi-structured interview questions were used to explore their reasons for enrolling and continuing in the program, the key takeaways they gained, and their suggestions for program improvement. Each interview lasted an hour and was audio recorded, professionally transcribed verbatim, and de-identified. The bilingual research assistant translated the Mandarin transcriptions into English, and the PI validated the accuracy and consistency of translation. Each participant received a gift card of 30 USD as compensation for their time.

### 2.3. Program Development and Description

The development of the program started with a needs assessment. During a local Chinese American community event, the PI announced the plan to initiate a community brain health education program and distributed a short questionnaire that invited Mandarin-speaking Chinese Americans 50 years or older to indicate their preferred meeting time and delivery mode. In addition, they were asked to suggest and rank brain health-related topics to be included in the education program. Among the 73 participants, Saturday was the preferred day of the week (64%). Although most participants preferred the in-person delivery mode, over a quarter (27%) of participants strongly preferred online participation. Using those responses, we scheduled almost all sessions on Saturdays and provided in-person and online delivery modes for each session. Participants’ ranks on topics of interest were utilized to design the curriculum and to identify appropriate instructors. For participants who struggled to identify topics of interest, the research team, based on the previous literature and national guidelines on dementia risk prevention [26,27,28,29,30], provided nine options for participants to consider, including community brain health resources, healthy diet, stress, physical activity, sleep, health checks, stress management, friendship and social support, basic knowledge of dementia, and financial management. On average, the highest-ranked topic was diet, followed by sleep, exercise, stress management, health checks, and basic knowledge of dementia. These preferences constituted the main topics for the six sessions of the brain health program.

An expert in each selected topic was invited to lead the corresponding session. The instructor must have had an advanced degree and possess research or clinical experience in the topic, identify as Asian (preferably Chinese ethnicity), and have prior experience providing community education or training for Asian populations (preferably Chinese ethnicity). The ability to communicate in Mandarin was also preferred. The final instruction team included a gerontological social work researcher specializing in dementia prevention and care (PhD and MSW), a social worker specializing in mindfulness and stress management (PhD candidate and MSW), a geriatric medical doctor with expertise in sleep improvement (MD), and a medical doctor and researcher specializing in nutrition and non-communicable chronic disease prevention and management (PhD and MD). Instructors were required to explain the connection between each topic and brain health and address questions and provide examples relevant to the older Chinese American population. As a key component of cultural tailoring, each instructor incorporated their practical experiences of working with Chinese or Mandarin-speaking Chinese Americans into the presented topics. For instance, the session on basic dementia knowledge addressed the stigma often found in Chinese communities. The session on healthy diets highlighted the nutritional values and culinary practices typical of the Chinese diet. Likewise, the mindfulness session included Qi Gong—a traditional Chinese mindfulness practice—to illustrate the concept of mindfulness for many participants who were unfamiliar with it.

Based upon the logical flow of topics and the availability of instructors, the program consisted of six sessions presented in the following order: (1) Alzheimer’s Disease and related dementias (ADRD), including its screening, diagnosis, treatment, and care, (2) nutrition and brain health among middle-aged and older adults, (3) sleep disturbance and dementia, (4) physical exercise and brain health, (5) interpretation of common medical examination reports and health guidance for middle-aged and older adults, and (6) meditation for better sleep and stress relief. Each session consisted of a 45–60 min presentation component, followed by a 10 min break and a 30–45 min “Q&A” component. The set-aside “Q&A” component is another important culturally tailored aspect of the program as many participants expressed strong preferences for having enough time to ask personalized questions. The final session also included a 30-min review. All lecture slides or notes were developed in Chinese, printed, and distributed to the in-person session participants and emailed to online participants.

In summary, the program was characterized by its responsiveness to community needs, tailored content addressing multiple components of brain health, and the involvement of qualified experts to ensure rigor and cultural relevance. Developed using a community-engaged approach, the program was informed by a needs assessment involving Mandarin-speaking Chinese Americans aged 50 and older. It offered flexible delivery modes and culturally relevant topics like diet, sleep, and exercise, guided by instructors with specialized experience and cultural competence. Sessions featured a mix of presentations and Q&A segments to facilitate interaction and address personal questions. Finally, critical measures of feasibility and effectiveness were included in program evaluation, as described below.

### 2.4. Measures

Outcomes included program feasibility, knowledge about risk factors including diet, exercise, health check, and brain health, depressive symptoms, sleep quality, behavioral motivation of lifestyle change for dementia risk prevention, and dementia worry. Program feasibility was measured by the percentage of participants who attended at least four of the six sessions. Due to the absence of existing instruments, the research team developed the instruments for knowledge scores of diets, exercise, health check, and brain health, following the established literature including national dementia risk prevention guidelines [26,27,28,29], program content, and cultural factors relevant to this population’s lifestyle. Survey questions and scoring procedures are included in the Appendix A.

Depressive symptoms were assessed by the 10-item Centre for Epidemiological Studies Depression (CES-D) scale [31]. Participants were asked to report the frequency of 10 statements in the past week, such as “I felt fearful” and “I had trouble keeping my mind on what I was doing”. Responses included “1 = less than 1 day”, “2 = 1–2 days”, “3 = 3–4 days”, and “4 = 5–7 days”. After reverse coding for applicable items, responses were summed so that a higher value indicates greater depressive symptoms (Cronbach’s α = 0.83).

Sleep quality was assessed using five items from the Boston Puerto Rican Health Study [32]. Example questions are “How frequently do you have difficulty falling asleep?”, “How frequently do you feel truly rested when you wake up in the morning?” Responses included “1 = most of the time”, “2 = sometimes”, and “3 = almost never or never”. After reverse coding for applicable items, response values were summed so that a higher value indicates better sleep quality (Cronbach’s α = 0.64).

Behavioral motivation of lifestyle changes for dementia risk prevention was adapted by seven items from the Motivation to Change Lifestyle and Health Behavior for Dementia Risk Reduction (MCLHB-DRR) scale [33]. The seven adapted items were summed, with higher scores indicating greater motivation to change. The scoring procedures are included in the Appendix A (Cronbach’s α = 0.83).

Dementia worry was measured by 10 items from the Dementia Worry Scale [34]. Using a 5-point scale, participants rated how typical each statement was of them (1 = not at all typical of me to 5 = very typical of me). The scores were totaled, with a possible range of 5–50. Higher values reflect greater levels of dementia worry (Cronbach’s α = 0.97).

In addition to outcome measures, we also collected demographic information, including age, sex, marital status (married/partnered or widowed/separate/single/divorced), education (below college degree or at least college degree), living alone (yes or no), number of children, and years lived in the United States (U.S.). All survey questions were translated into Mandarin by one researcher and validated by another researcher on the team.

### 2.5. Data Analysis

Data analysis followed a convergent parallel mixed-methods framework. For quantitative data, frequency and percentage were used to summarize the attendance and describe the program’s feasibility. Descriptive statistics, including mean and standard deviation for continuous variables and percentage for categorical variables, were used to summarize sample characteristics for all participants who registered for the program, as well as by subgroup based on completion status. Independent-sample *t*-tests and chi-squared tests were used to examine whether sample characteristics significantly differed between those who completed the program and those who did not. This analysis helped identify potential factors contributing to participant attrition. Paired-sample *t*-tests were used to compare all outcomes’ pre-test and post-test scores, which allows us to obtain preliminary findings regarding the program’s effectiveness on the above-mentioned outcomes. All quantitative data analyses were conducted using Stata 18 MP.

For qualitative data, thematic analysis was conducted to validate quantitative findings regarding the program’s feasibility and effectiveness [35]. The analysis was led by two bilingual co-authors under the supervision of the PI. Inconsistencies in thematic coding were resolved after a discussion between the two coders and the PI. The analytical process unfolded in four stages to enhance rigor. First, all researchers independently reviewed the feedback questionnaires and focus group transcripts to familiarize themselves with the raw data. Second, the two lead analysts employed an open coding approach, individually identifying codes and developing the initial codebook. Third, the research team convened multiple times to synchronize and refine the codebook until a consensus was reached on the coding scheme. Fourth, using the refined codebook, each analyst revisited the transcripts and individually coded the data using the agreed-upon scheme. At the same time, analysis teams held regular meetings to discuss and resolve any discrepancies. Finally, themes and subthemes were validated among the full team, and exemplar quotes for each theme were selected and translated into English. MAXQDA 24.5.1 was used to facilitate qualitative data analysis.

In line with the convergent mixed-methods framework, the integration of quantitative and qualitative findings was conducted after completing the separate analyses. Quantitative results were compared with the themes identified through qualitative analysis. The research team triangulated the data by identifying areas of convergence where both data types aligned, such as the program’s effectiveness and feasibility, while also exploring discrepancies to generate a more nuanced interpretation. This process ensured that both numerical trends and participant narratives were synthesized to provide a comprehensive understanding of the program’s impact.

## 3. Results

### 3.1. Quantitative Results

#### 3.1.1. Feasibility

Figure 1 illustrates the feasibility of the study, as indicated by attendance. In total, 77 participants, including 62 in-person and 15 online, registered for the program and consented to the research study. Almost 90% (*n* = 69) of all research participants achieved the feasibility endpoint by attending at least four out of six sessions. The in-person and online group completion rates were 94% (*n* = 58) and 73% (*n* = 11), respectively. The results of the two-sample test of proportions showed that the completion rate for the in-person group was significantly higher than that of the online group (z = 2.30, *p* = 0.02).

#### 3.1.2. Sample Characteristics

Table 1 displays a summary of the sample characteristics for the whole sample (Column 1) and by completion status (Column 2 and Column 3). The test statistics and corresponding *p*-values comparing the sample characteristics between the two completion status groups are presented in Column 4 and Column 5, respectively. To highlight a few, the average age of the whole sample was 75.96 years (SD = 10.52). More females (73%) than males (27%) registered in the program. Most participants were married or partnered (69%), and a total of 29% lived alone. The average number of children was 1.75 (0.96). Over half of the participants (54%) had an education below a college degree. On average, participants had lived in the U.S. for 20.80 years (SD = 11.31). There was no statistical difference in any baseline characteristic between those who completed the program and those who did not.

#### 3.1.3. Paired-Sample *t*-Tests

Table 2 displays the results of paired-sample *t*-tests to summarize preliminary findings regarding the program’s effectiveness. Compared to the pre-test, significant increases were observed in sleep quality (*t* = 2.43, *p* = 0.019), behavioral motivation of lifestyle changes (*t* = 2.20, *p* = 0.033), and knowledge of diet (*t* = 2.50, *p* = 0.016), exercise (*t* = 2.34, *p* = 0.024), health check (*t* = 3.37, *p* = 0.002), and brain health (*t* = 2.26, *p* = 0.029), with Cohen’s d ranging from 0.33 to 0.50. In addition, depressive symptoms significantly decreased (*t* = −2.33, *p* = 0.025), with Cohen’s d being −0.35. No significant change was observed for dementia worry (*t* = −1.52, *p* = 0.137, Cohen’s *d* = −0.23).

### 3.2. Qualitative Findings

The thematic analysis of the qualitative data revealed three major themes: (1) *promising feasibility and acceptability*, (2) *perceived benefits*, and (3) *areas for improvement* as reported in the session feedback questionnaires or focus groups. Overall, the qualitative analysis found that well-designed, accessible health education programs can effectively empower older adults with knowledge and strategies for maintaining brain health, potentially contributing to positive lifestyle changes and improving their quality of life.

#### 3.2.1. Promising Feasibility and Acceptability

The promising feasibility and acceptability of the workshops can be seen from these three key aspects: (1) *appropriate timing and duration*, (2) *a comprehensive and interactive delivery structure*, and (3) *provision of scaffolding*.

**Appropriate timing and duration.** Participants consistently reported that the timing and duration of each workshop were well-planned and conducive to attendance. One participant articulated this sentiment:


*“I think the timing of the sessions was well planned. Firstly, it accommodated our location needs. If these were held on weekdays, we wouldn’t have been able to secure this venue. Secondly, your staff wouldn’t have been available due to work commitments… So, I think the length and timing of the sessions were quite well arranged”.*


This quote represents the dual benefits of the chosen schedule, addressing logistical constraints and aligning with participant and speaker availability. Another participant noted, “*Utilizing Saturday mornings was a very good choice*”. These comments suggest that scheduling based on stakeholder feedback played a crucial role in making the program more accessible, which consequently increased its feasibility and acceptability among participants.

**Comprehensive and interactive delivery structure.** Participants reported that the workshop structure was well designed, particularly noting the inclusion of discussion sections. This format reportedly facilitated better understanding and engagement and allowed participants to process the content more effectively. One participant shared:


*“The structure you arranged has very strong logic. It’s tailored to human developmental needs, allowing immediate recall and response. So, it’s a very natural and effective approach”.*


This statement highlights the perceived effectiveness of the workshop structure, emphasizing its logical organization and alignment with individual learner’s cognitive processes. The participant’s use of terms like “immediate recall and response” suggests that the structure facilitated active learning and engagement.

Furthermore, including discussion time enhances the workshops’ acceptability and effectiveness by allowing participants to engage with the material more deeply. As another participant pointed out that “*the discussion also helps us recall the lecture’s content and summarize*, *so it’s quite beneficial*”. This interactive element likely contributed to the program’s effectiveness by enabling participants to clarify concepts, share experiences, and reinforce their learning.

**Provision of scaffolding.** Considering the unique needs of the older Chinese American population, the workshops incorporated various scaffolding techniques. These included providing hardcopy materials, adapting the instructors’ speaking pace with repetition when necessary, and utilizing real-life examples to facilitate comprehension. Participants reported that these scaffolding methods greatly enhanced the workshops’ acceptability and effectiveness. One participant highlighted the value of the printed materials: “*What I liked the most about this lecture series is that we were given handouts*, *which allowed us to review the content. We can often refer to them*”. This statement underscores the importance of tangible resources in supporting ongoing learning and retention among older adults. Another participant supported this view, emphasizing the relevance for those with age-related sensory changes: “*The lecture notes (handouts) are very important*, *especially for older people whose hearing may not be as good. Having the notes means they will naturally refer to them*”. By accommodating potential auditory limitations and providing a means for review and reinforcement, these scaffolding techniques significantly contributed to the program’s overall effectiveness.

#### 3.2.2. Perceived Benefits

Participants reported benefiting from the workshops in three primary ways: (1) *increased awareness of preventive care*, (2) *enhanced health literacy*, and (3) *decreased anxiety/fear*.

**Increased awareness of preventative care.** The analysis revealed that participants strongly intended to modify their lifestyles following the workshops to maintain brain health. This included the willingness to make dietary changes, undergo regular health check-ups, exercise regularly, improve sleep quality, and maintain social activities. Notably, some participants reported significant shifts in their lifestyles to prevent dementia. For example, one participant described how the workshops helped change her eating habits:


*“I didn’t take it seriously until the lecture pointed out the high-risk factors for dementia. I realized that continuing my old habits could lead to a burden on my children and society. So now, I’ve started to be more cautious. Although I haven’t quit sugar completely, as I love desserts, I’ve cut down significantly, even drinking only black coffee”.*


Additionally, another participant shared the transition she had in her life to prevent dementia:


*“After attending this lecture series, I’ve become very proactive. I started wondering how to prevent dementia…I’ve signed up for an online piano class. Every day, I wake up at 3 a.m. to attend the class, and practicing piano has greatly helped my cognitive functions and provided musical enrichment”.*


These accounts demonstrate how the workshops successfully raised participants’ awareness of brain health and motivated participants to adopt proactive and meaningful lifestyle changes aimed at reducing dementia risk, highlighting the program’s positive impact on preventative health behaviors.

**Enhanced health literacy.** The analysis yielded that, upon completion of the workshops, most participants reported an improvement in their health literacy, such as a better understanding of nutritional concepts, increased scientific knowledge about specific dietary components (such as proteins and fatty acid types), and a greater appreciation for the importance of dietary diversity. For example, one participant shared: “Eating certain food is helpful for sleep, such as food that contains amino acid, vitamin Bs, and fish oil, etc.”. Furthermore, through attending seminars, some participants critically evaluated and corrected their existing misperceptions about health and nutrition, and one participant stated:


*“My perspectives on diet have changed significantly. Before, I used to think that eating less or not eating would lower my blood sugar or help me lose weight, but these ideas were baseless. After listening to these systematic health lectures, I realize the importance of a proper diet for my health”.*


Additionally, the enhanced health literacy seemed to benefit their family caregiving for parents, as noted by one participant:


*“Understanding how to read various parts of medical reports systematically has been beneficial, especially considering my elderly parents’ health. I talk to my parents every day and monitor their health status. I use the knowledge I’ve gained to care for them daily, which has been very useful”.*


Participants also successfully transferred the knowledge into practice. One participant shared a successful experience of practicing mindfulness learned in this program to greatly improve sleep quality:


*“The last class taught about ‘breathing in blue air and exhaling red air.’ My sleep has improved dramatically; it’s almost unbelievable. I’ve been seriously adhering to it, … This has been the best state of sleep I’ve ever experienced”.*


This account illustrates how integrating mindfulness techniques into daily life can lead to tangible improvements, such as enhanced sleep quality, showcasing the practical value of the program.

**Decreased worry.** Even though our quantitative study did not show a statistically significant decrease in dementia worry, qualitative results indicated that the workshops also reduced participants’ anxiety about dementia, positively impacting their overall well-being. A key factor in this reduction was the improved ability to distinguish between normal age-related cognitive decline and dementia symptoms. One participant eloquently expressed this shift in her perspective:


*“I’m not as worried about dementia anymore. I ask if I show signs of it whenever I see a doctor. My daughter also points out how I’m always worried about dementia. But now I understand that it’s natural for cognitive functions to decline with age, and it’s important to manage one’s thoughts and not be scattered, which can worsen everything. This understanding has been very helpful”.*


This quote demonstrates how increased knowledge led to a more balanced view of cognitive aging, reducing fear and promoting a more proactive approach to brain health. The workshops have empowered participants with information that allowed them to contextualize their experiences and concerns within a deeper understanding of cognitive aging.

#### 3.2.3. Areas for Improvement

Three major areas for improvement were suggested by participants: (1) incorporating bilingual study materials, (2) including more experience-sharing components, and (3) adding more time for “Q&A”.

**Incorporating bilingual study materials**. For both session feedback questionnaires and focus group interviews, some participants suggest that incorporating bilingual study materials would have had allowed them to better understand their medical reports.


*“For someone like me, who reads medical reports entirely in English, encountering a Chinese medical term can be confusing. I don’t always know how to match those terms with the English terms in my medical reports… it would be helpful if medical terms were presented in both English and Chinese. This way, when we look at our English medical reports or blood test results, we can easily make the connection”.*


This feedback highlights the importance of bilingual materials in bridging language barriers and facilitating better comprehension of medical information.

**Including peer experience-sharing components.** Several participants expressed a strong interest in adding experience-sharing components through which participants can establish peer support and learn from each other. For example, one participant stated:


*“It would be great if we could exchange our experiences and add to what the teachers discussed with actual results. For instance, someone mentioned diabetes. I don’t have diabetes, but my A1C is in the prediabetes range, so I pay attention to that. I believe Chinese diets often include too many refined carbohydrates, which should be avoided. I can share my experience with that. And if someone can sleep through the night, they could share their experience. If time allows for such exchanges, it would be even better”.*


Another participant added:


*“We really need a support system. I hope that through these lectures, I can connect with other direct caregivers or those with family members suffering from the disease, so we can exchange experiences and support each other”.*


One more participant supported the statement and expressed a strong desire to share their personal experience and “even serving as an educator to help and inform” others. Those responses underscore participants’ enthusiasm for peer experience-sharing to build a support network and deepen their learning.

**More time for “Q&A”.** Several participants suggested longer time for “Q&A” to clarify details for important concepts and better remember the concepts. One participant, for instance, stated:


*“I feel it would be better if there was more time for discussion. The instructor’s lectures are informative, and we have questions or points of interest as he speaks. Sometimes we might read about these topics, but it’s not the same as remembering them. The lectures leave a deeper impression. So, having more time to discuss and delve into the topics would be beneficial”.*


This suggestion highlights the value participants place on interactive learning and their desire for opportunities to engage more deeply with the material presented.

## 4. Discussion

Building on the existing literature and prior research in brain health education, this study is among the first to specifically address the knowledge gap in delivering and evaluating structured community education focused on brain health to Chinese Americans aged 50 years or older. The primary goal of this study is to examine the feasibility of a culturally tailored community brain health education program with a mixed-methods approach. The findings provide promising evidence for the feasibility of this program, with about 90% of participants completing the program (i.e., attending four or more out of six sessions).

The qualitative findings not only confirmed the quantitative findings but also revealed three major elements contributing to the perceived feasibility by participants, including (1) appropriate timing and duration, (2) comprehensive delivery structure incorporating discussion, and (3) provision of scaffolding. First, appropriate timing greatly enhanced the program’s feasibility, highlighting the importance of assessing participants’ availability. In our program, most participants had other activities scheduled on weekdays and preferred to spend time with family members on Sundays. Additionally, many of our staff and volunteers were only available on weekends to support preparing for the in-person sessions. Thus, careful planning through needs assessments helped us maximize participants’ engagement and community resources. In addition, it is important to recognize the attention span of older adults when determining the duration of each session. Previous successful health literacy education programs for older adults, according to a systematic review, limited each session to 1–2 h [36]. Thus, we limited each session to 1.5 h and planned at least a 10-minute break within each session to ensure that participants could stay engaged throughout each session.

Second, the delivery structure was another critical element influencing perceived feasibility and acceptability by participants, particularly the inclusion of discussion sections. Participants highlighted the effective organization/structure of the workshops, which facilitated active learning and engagement. Our session structures, described by one participant as “tailored to human developmental needs”, promoted immediate recall and response, aligning well with cognitive processes. The inclusion of discussion sections further enriched the learning experience by providing opportunities for participants to clarify, process, and reinforce the content collaboratively while also receiving personalized input/feedback. This tailored approach ensured that each participant felt heard and supported, enhancing their motivation to apply the knowledge and strategies discussed in the workshops. Collectively, these elements created a dynamic and supportive learning atmosphere that maximized the impact of the program and empowered participants to take proactive steps toward improving their brain health.

Third, the incorporation of scaffolding techniques is instrumental in enhancing the feasibility and effectiveness of health education [37]. The use of hardcopy materials, such as handouts, proved particularly valuable, as highlighted by participants who appreciated the ability to review and reinforce learning at their own pace. Moreover, by addressing the unique needs of this population, such as age-related sensory changes, language barriers, and the need for repetition, the lecture notes and summary handouts effectively facilitated comprehension and engagement. These features cater to the needs of older learners by fostering an inclusive and collaborative learning environment.

The secondary goal of this study is to evaluate the program’s effectiveness on brain health-related knowledge, behaviors, short-term outcomes, and dementia worry. The findings show that participants’ knowledge (including diet, exercise, health check, and general brain health) and behavioral motivation for lifestyle changes have significantly improved, aligning with the effects observed in previous community brain health education programs [10,11,12]. These positive changes were further corroborated by qualitative data on increased awareness of preventive care and health literacy. Although the current study design does not lead to definitive evidence of a causal relationship between the program and short-term health outcomes, both the quantitative and qualitative findings suggest positive short-term health effects. Specifically, improvements in sleep quality and reductions in depressive symptoms were observed when practical techniques were explained and demonstrated as part of this community health education. While our qualitative data alluded to the program’s impact on reducing dementia worry, our quantitative results did not show significant changes. Dementia worry often stems from deeply ingrained fears, beliefs, and cultural stigma surrounding the condition [38,39]. The program duration may not have been sufficient to induce measurable changes in deeply rooted fears and beliefs about dementia, highlighting the need for future educational efforts to include longer-term engagement and alter their perceptions of dementia. Furthermore, cultural factors may play a crucial role in shaping attitudes toward dementia, with some participants perhaps reluctant to acknowledge minor improvements or wary of expressing reduced worry openly [40]. This underscores the importance of tailoring educational programs to recognize and address these deep-seated cultural aspects.

The results on areas for improvement provided valuable insights to strengthen the feasibility and effectiveness of community brain health education programs. Incorporating bilingual study materials is critical for allowing participants to understand the educational content, utilize community resources, and communicate with healthcare providers. Additionally, participants expressed a strong desire for experience-sharing opportunities to establish peer support and learn from others’ practical strategies for managing health. These peer exchanges could greatly reinforce self-regulation and skill-building activities, foster community, and provide participants with actionable insights to achieve health goals [41]. We consider this suggestion an important area for future programs. Finally, extending the “Q&A” session was suggested to address participants’ specific questions, reinforce important concepts, and deepen understanding through discussion. Together, these recommendations underscore the importance of tailoring community education programs to meet participants’ diverse needs and preferences, ultimately enhancing their engagement and the program’s overall impact.

Several limitations should be acknowledged. First, this study used a single-arm pre–post-test design, which did not compare participants with a control condition or an active comparison. Thus, findings regarding the program’s effectiveness remain preliminary. A randomized controlled trial design is warranted to address potential confounders and test the assumptions of causality. Second, this study used convenience sampling with a relatively small sample size of older adults, most of whom lived in senior housing facilities, which may have limited the generalizability of findings. In addition, participants who chose to join the focus groups may feel more engaged in the program than those not in the focus groups. Thus, using a convenience sample for focus groups may have prevented collecting in-depth information about factors that hindered engagement in the program. Finally, we acknowledge the limitation that our custom-developed instruments for assessing knowledge do not yet have fully established psychometric properties, particularly for this population. This highlights the need for future research to develop and rigorously test these instruments to ensure their reliability and validity among Mandarin-speaking Chinese Americans.

In conclusion, this study, to our knowledge, is the first to use mixed methods to demonstrate the promising feasibility and potential effectiveness of structured and culturally tailored community brain health education for Chinese Americans. The findings highlight the promise of such programs in improving brain health-related knowledge, behaviors, and short-term health outcomes, particularly when incorporating culturally relevant content and interactive elements. While further research with more rigorous designs is needed, this study, with both quantitative and qualitative data on the program’s strengths and limitations, lays a foundation for developing and implementing future community-based interventions to address brain health and dementia risk prevention for older Chinese Americans and other underserved populations.

## Figures and Tables

**Figure 1 geriatrics-10-00058-f001:**
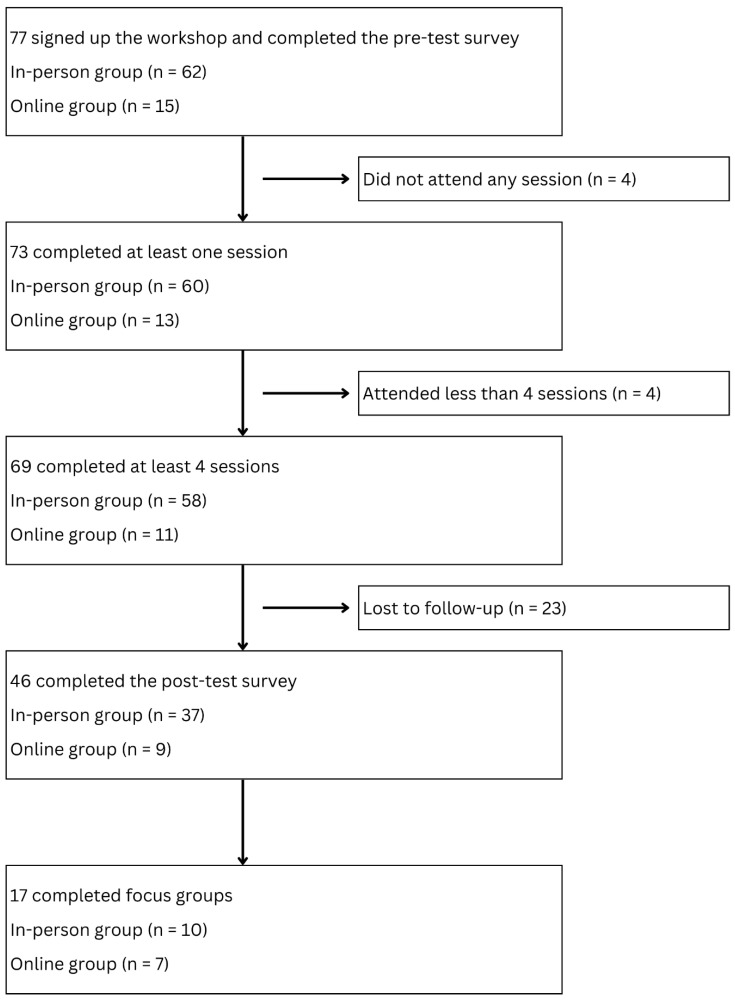
Participant flow chart.

**Table 1 geriatrics-10-00058-t001:** Sample characteristics for the whole group and by completion status.

	Whole Group (*n* = 77)	Completed (*n* = 69)	Not Completed (*n* = 8)	t or χ^2^ Statistic	*p*-Value
Age in years	75.96 (10.52)	75.84 (10.04)	76.88 (14.71)	0.26	0.796
Sex				2.32	0.127
Male	27%	25%	50%		
Female	73%	75%	50%		
Marital status					
Married/partnered	69%	70%	64%	0.17	0.683
Widowed/single/separate/divorced	31%	30%	37%		
Number of children	1.75 (0.96)	1.78 (0.98)	1.50 (1.01)	−0.64	0.523
Living alone				2.01	0.156
Yes	29%	26%	50%		
No	71%	74%	50%		
Education				0.97	0.324
Below college degree	54%	56%	38%		
College degree or above	46%	44%	62%		
Years lived in the United States	20.80 (11.31)	20.34 (11.31)	24.25 (15.21)	0.88	0.381
Knowledge: diet	3.96 (1.68)	4.00 (1.66)	3.63 (1.92)	−0.59	0.554
Knowledge: exercise	2.66 (1.71)	2.71 (1.67)	2.25 (2.05)	−0.72	0.473
Knowledge: health check	3.16 (2.01)	3.17 (1.97)	3.00 (2.45)	−0.23	0.818
Knowledge: brain health	7.96 (2.42)	7.93 (2.51)	8.25 (1.58)	0.35	0.724
Depressive symptoms	19.84 (5.85)	19.96 (5.75)	18.86 (7.02)	−0.85	0.624
Sleep quality	10.14 (2.12)	10.04 (2.13)	11.00 (1.93)	1.21	0.228
Behavioral motivation	24.14 (4.78)	24.16 (4.75)	23.94 (5.43)	−0.12	0.908
Dementia worry	20.24 (12.13)	20.56 (12.51)	16.83 (6.68)	−0.72	0.476

Note: “Completed” refers to those who attended at least four sessions, while “Not completed” refers to those who attended less than four sessions.

**Table 2 geriatrics-10-00058-t002:** Results of Paired-sample *t*-tests (*n* = 46).

Outcomes	Pre-Test Mean (SD)	Post-Test Mean (SD)	Paired Mean Difference [95% Confidence Interval]	*t*-Statistic	*p*-Value	Cohen’s *d*
Knowledge: diet	4.28 (1.76)	4.89 (1.34)	0.61 [0.12, 1.10]	2.50	0.016	0.37
Knowledge: exercise	2.93 (1.73)	3.48 (1.64)	0.54 [0.08, 1.01]	2.34	0.024	0.35
Knowledge: health check	3.28 (1.92)	4.48 (1.95)	1.20 [0.48, 1.91]	3.37	0.002	0.50
Knowledge: brain health	7.91 (2.76)	8.76 (1.30)	0.85 [0.09, 1.60]	2.26	0.029	0.33
Mood and feelings	18.67 (5.13)	17.03 (3.63)	−1.64 [−3.06, −0.22]	−2.33	0.025	−0.35
Sleep quality	10.24 (2.05)	10.76 (2.23)	0.52 [0.09, 0.95]	2.43	0.019	0.37
Depressive symptoms	24.66 (5.25)	26.53 (4.37)	1.88 [0.16, 3.59]	2.20	0.033	0.33
Dementia worry	16.56 (8.48)	14.66 (5.17)	−1.90 [−4.44, 0.53]	−1.52	0.137	−0.23

## Data Availability

Data is not available due to privacy and ethical restrictions.

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
