# Peer review of "Culturally Tailored Community Brain Health Education for Chinese Americans Aged 50 or Above: A Mixed-Methods Open Pilot Study"

_geriatrics, 2025, doi:10.3390/geriatrics10020058_

Round 1

Reviewer 1 Report

Comments and Suggestions for Authors

Ref.: geriatrics-3445680

Although being a pilot study, this is an interesting work, dealing with education of the community on brain health issues, especially important for the prevention of dementia. Genetic risk factors cannot be changed; however modifiable risk factors may be very important in reducing the risk of cognitive decline. Thus, such studies are welcomed worldwide and stress the need of community-based education.

The authors developed and evaluated a culturally tailored community brain health education program, aiming to enhance brain health knowledge and motivate lifestyle changes among Chinese Americans aged 50 or older, which (based on current knowledge) are expected to reduce the risk of dementia.

The authors included practically all modifiable risk factors, including dietary parameters, physical exercise, health check, brain health parameters, mood, depression and sleep. Feasibility and acceptability of the education program were high and, following the program, significant improvements in brain health knowledge, sleep quality, behavioral motivation for lifestyle changes and depressive symptomatology were noted.

The study is well designed-executed and well written. Since dementia is a major health problem worldwide, the present study stresses the need for community-based educational programs not only in specific subpopulations but across all ethnic groups.

I have no specific comments. I believe that if and when published, this paper will receive many citations.

Author Response

The reviewer did not provide specific comment, yet graciously affirmed the importance of our study. We deeply appreciate the thoughtful and supportive review, especially the reviewer's positive feedback on our study's design, execution, and significance. Such recognition of the importance of community-based brain health education further reinforces our commitment to advancing dementia prevention efforts, especially among older adults in the marginalized communities.

Reviewer 2 Report

Comments and Suggestions for Authors

You can see my comments below.

Abstract

Background should include the very brief introduction about the research area and target population then the knowledge gap. 

Only mentioned 4 phases simply. Then you can have more space to the outcome measures and analytic results.

Results should include analytic findings, eg. paired t test and its significance...

Introduction

The linkage of 1st and 2nd paragraphs is unclear. How dementia and then lifestyle factors for brain health and related program link together is unclear.

What is brain health?

What are the characteristics of the education program? How effectiveness of the program in the past studies? what outcome measures were examined using the program?

In the past studies, the programs could be different. In your study, what were the characteristics of each section in your study?

The outcomes should be explained, esp in this target population. Why were this population received the program had the assessment of those outcomes?

Materials and methods

participants: any exclusion criteria?

Better to separate the development of the program

Data analysis:

who was the interviewer? how to conduct the interviews? the interviews were recorded? How long were the interviews? Who transcribed to the verbatim? what method was used to analyse the script? The themes eventually translated into English. I assume that the dialogues related to the themes were translated. Any member check to ensure the translation was correct and consistent with the meaning?

Results:

Table 1 indicates 2 group, completed and incompleted. I wonder why the groups were not received and not received the program? what was the meaning for the t-test between them?

Discussion

"This project, to our knowledge, is among the first published studies' must be carefully written in your paper.  Your paper was to address the knowledge gap that may be in different population or age groups. The designs and the outcome measures should be taken as references from the past papers. 

Author Response

Comment 1: Abstract: Background should include the very brief introduction about the research area and target population then the knowledge gap. 

Response 1: Thank you. We accept the comment and add the information in our revised abstract.

Comment 2: Only mentioned 4 phases simply. Then you can have more space to the outcome measures and analytic results.

Response 2: Thank you for your comment. Fortunately, we have enough space to include all the outcome measures and analytic results without cutting down the phrases.

Comment 3: Results should include analytic findings, e.g. paired t test and its significance...

Response 3: Thank you for your comment. We accept the comment and specified statistical test (paired t-test) and clarified that two-tailed p-values were all smaller than 0.05 in our revised abstract.

Comment 4: The linkage of 1st and 2nd paragraphs is unclear. How dementia and then lifestyle factors for brain health and related program link together is unclear. What is brain health?

Response 4: Thank you for your comment. We accept the comment and provide the link and the brain health’s definition in the revised manuscript.

Comment 5: What are the characteristics of the education program? How effectiveness of the program in the past studies? what outcome measures were examined using the program? In the past studies, the programs could be different. In your study, what were the characteristics of each section in your study?

Response 5: We appreciate the reviewer's interest in the specifics of our education program. In the revised manuscript, we have clarified the key characteristics of our program, which include responsiveness to community needs, culturally relevant content covering multiple aspects of brain health, and expert-led sessions to ensure academic rigor and relevance. As noted, our program is innovative and fills a gap in existing literature, as we found no prior programs directly comparable to ours after a thorough literature search.

Comment 6: The outcomes should be explained, esp. in this target population. Why were this population received the program had the assessment of those outcomes?

Response 6: Thank you for your comment. As highlighted in our revised manuscript, we carefully developed the instruments for knowledge scores of diet, exercise, health checks, and brain health by following established literature, including national dementia risk prevention guidelines and cultural factors specific to our target population's lifestyle (21–24). By aligning our outcome measures with these guidelines, as well as the program content and cultural factors relevant to this population’s lifestyle, our intent was to ensure the program's relevance.

Comment 7: Materials and methods: participants: any exclusion criteria?

Response 7: Thank you for your comment. There are no exclusion criteria applicable to this study.

Comment 8: Better to separate the development of the program

Response 8: Thank you for your comment. Participants and the development of the program were explained in different sections. Participants, including eligibility criteria, are described in Section 2.1, whereas the development of the program is described in Section 2.3.

Comment 9: Who was the interviewer? how to conduct the interviews? the interviews were recorded? How long were the interviews? Who transcribed to the verbatim? what method was used to analyse the script? The themes eventually translated into English. I assume that the dialogues related to the themes were translated. Any member check to ensure the translation was correct and consistent with the meaning?

Response 9: These are important questions. In the revised manuscript, questions related to the interviews, interviewers, translations, and validations have been further clarified, which is highlighted in the last paragraph of Section 2.2 (Design Overview and Study Procedures).  For the question regarding analysis method, we used the thematic analysis approach. More detailed responses for the analysis method is described in the second paragraph of Section 2.5.

Comment 10: Table 1 indicates 2 group, completed and incompleted. I wonder why the groups were not received and not received the program? what was the meaning for the t-test between them?

Response 10: Thank you for your question. We added a footnote to clarify “completed” vs “not completed” in the revise manuscript, consistent with Section 2.5. All participants received the program, yet the degree of receipt depends on the number of sessions attended. Therefore, “completed” and “not completed” are the more accurate terms here.  This analysis is important because it helped identify potential factors contributing to participant attrition, which we added to Section 2.5. We acknowledge the importance of understanding the importance of reasons for not receiving. However, this is beyond the scope of a pilot study that is designed to be a single-group pretest-posttest design.

Comment 11: Discussion "This project, to our knowledge, is among the first published studies' must be carefully written in your paper.  Your paper was to address the knowledge gap that may be in different population or age groups. The designs and the outcome measures should be taken as references from the past papers. 

Response 11: Thank you for the suggestion. We accepted this comment. This language has been revised per the reviewer’s feedback as “Building on existing literature and prior research in brain health education, this study is among the first to specifically address the knowledge gap in delivering and evaluating structured community education focused on brain health to Chinese Americans aged 50 years or older.”

Reviewer 3 Report

Comments and Suggestions for Authors

The ms. titled “Culturally Tailored Community Brain Health Education for Chinese Americans Aged 50 or Above: A Mixed-Methods Open Pilot Study” aims to evaluate a culturally tailored community brain health education program to enhance brain health knowledge and motivate lifestyle changes to prevent risks of dementia among Chinese Americans aged 50 or older. Quantitative results showed significant increases in brain health knowledge, sleep quality, behavioral motivation for lifestyle changes, and decreased depressive symptoms. Qualitative results further revealed the high feasibility, acceptability, and perceived benefits of the program. This ms. try to improve the clarity and knowledge of the topic to promote brain health and lifestyle changes.

The work is well written in the English language, and it is easy to follow. Anyway, in my opinion, some minor changes are needed to enhance the quality of the ms.

INTRODUCTION

1)    The introduction could better contextualize the significance of addressing modifiable risk factors in the broader framework of public health interventions. I suggest the authors address this point.

2)    It would benefit from a more detailed explanation of the rationale behind focusing specifically on Mandarin-speaking Chinese Americans in the Denver area.

3)    I suggest explaining more accurately the aim of the study in a separate section

4)    The following query pertains to a theoretical aspect of the text. The paper makes reference to the perception and improvement of the quality of mental health among individuals aged 50 and above. However, the introduction makes note of the prevalence of dementia in populations over 65. The rationale behind considering individuals aged 50 and above, as opposed to 65, remains unclear. It is possible that the reason is not fully apparent, and further elucidation would be appreciated.

MATERIAL AND METHODS

5)    Some instruments, such as those assessing brain health knowledge, were developed by the research team, but their psychometric properties (e.g., reliability and validity) are not well-documented. I suggest the authors argue this choice. 

6)    I suggest moving the “Sample Characteristics” in the material and methods section to the first sub-paragraph.

7)    The description of the statistical methods and data analyses could be more explicit.

RESULTS

8)    Some tables lack a clear link to the narrative text, requiring readers to infer connections between the data and the study's objectives.

9)    The discussion would benefit from a more critical analysis of why certain outcomes, such as dementia worry, did not improve significantly.

DISCUSSION

10) The lack of significant change in dementia worry is mentioned but not adequately discussed, leaving an important finding underexplored.

11) The discussion would benefit from a more critical analysis of why certain outcomes, such as dementia worry, did not improve significantly.

12) In conclusion, it is imperative that the manuscript prioritize the concerns that are of relevance to the elderly population. To that end, it is recommended that this aspect be addressed with greater precision and specificity in this section. Please, consider to examine and incorporate these articles in the reference list: (https://doi.org/10.3390/ijerph19053097; 10.1016/S2468-2667(20)30185-7; 10.1016/j.archger.2016.02.007)

Author Response

We have addressed all the comments provided below. Please note that the original Comment 9, Comment 10, and Comment 11 from the reviewer were overlapping. Therefore, we have merged them into a single Comment 9 in our response and adjusted the numerical sequence according in our response below.

Comment 1: The introduction could better contextualize the significance of addressing modifiable risk factors in the broader framework of public health interventions. I suggest the authors address this point.

Response 1: Thank you for your comment. We accept this suggestion and better contextualized the significance of addressing modifiable risk factors in the broader framework of public health interventions in Paragraph 2 in the revised manuscript.

Comment 2: It would benefit from a more detailed explanation of the rationale behind focusing specifically on Mandarin-speaking Chinese Americans in the Denver area.

Response 2: Thank you for your comment. We accept this suggestion and provide a more detailed explanation in the first paragraph of Section 1.2 in our revise manuscript.

Comment 3: I suggest explaining more accurately the aim of the study in a separate section

Response 3: Thank you for your comment. We accept this suggestion and create a separate section (Section 1.2 in the revised manuscript) to more thoroughly explain the study aims and rationales.

Comment 4: The following query pertains to a theoretical aspect of the text. The paper refers to the perception and improvement of the quality of mental health among individuals aged 50 and above. However, the introduction makes note of the prevalence of dementia in populations over 65. The rationale behind considering individuals aged 50 and above, as opposed to 65, remains unclear. It is possible that the reason is not fully apparent, and further elucidation would be appreciated.

Response 4: Thank you for your comment. We accept this suggestion and provide further education to justify the age criterion based on the AARP recommendation in Section 1.2 of the revised manuscript.

Comment 5:   Some instruments, such as those assessing brain health knowledge, were developed by the research team, but their psychometric properties (e.g., reliability and validity) are not well-documented. I suggest the authors argue this choice. 

Response 5: Thank you for your comment. The instruments were developed by the research due to lack of instrument for the overall older population as well as Chinese Americans. We (1) provided the reason in Section 2.4 and (2) acknowledged this as a limitation in Section 4 (second last paragraph) of the revised manuscript.

Comment 6: I suggest moving the “Sample Characteristics” in the material and methods section to the first sub-paragraph.

Response 6: Thank you for your comment. We have reviewed similar articles published in this journal and found most of them reported sample characteristics in the results section (e.g., https://doi.org/10.3390/healthcare13020122, https://doi.org/10.3390/ijgi13120454). Therefore, we decided to keep “Sample Characteristics” in the results section for consistency.

Comment 7: The description of the statistical methods and data analyses could be more explicit.

Response 7: Thank you for your comment. We accept this suggestion and provide the specific statistical tests and clarified the purpose of each test in Section 2.5 of the revised manuscript.

Comment 8:  Some tables lack a clear link to the narrative text, requiring readers to infer connections between the data and the study's objectives.

Response 8: Thank you for your comment. We accept this suggestion and provide the link between each table and its purpose. This manuscript includes a total of two tables. For Table 1, we revised the first two sentences of Section 3.1.2 to clarify the link. For Table 2, we revised the first sentence of Section 3.1.3 to clarify the link.

Comment 9: The discussion would benefit from a more critical analysis of why certain outcomes, such as dementia worry, did not improve significantly. The lack of significant change in dementia worry is mentioned but not adequately discussed, leaving an important finding underexplored.

Response 9: Thank you for your comment. We accept this suggestion and provide additional discussion related to findings on dementia worry. More details can be seen at the end of the paragraph starting with “The secondary goal of this study …” in Section 4.

Comment 10: In conclusion, it is imperative that the manuscript prioritize the concerns that are of relevance to the elderly population. To that end, it is recommended that this aspect be addressed with greater precision and specificity in this section. Please, consider examining and incorporate these articles in the reference list: (https://doi.org/10.3390/ijerph19053097; 10.1016/S2468-2667(20)30185-7; 10.1016/j.archger.2016.02.007)

Response 10: We appreciate the reviewer's suggestion to examine and consider incorporating additional articles into our reference list. Upon reviewing the recommended articles, we found that while they offer valuable insights into issues relevant to risk factors among older adults in other countries, they do not directly pertain to or focus on Chinese Americans or culturally tailored interventions, which are central to our study's objectives. Our manuscript prioritizes concerns specific to the elderly Chinese American population, and we have ensured that our references reflect this focus.

Reviewer 4 Report

Comments and Suggestions for Authors

Authors used an interview survey to examine feasibility and acceptability of a community brain health education program. However, this article has not fully answered some of the questions due to insufficient description.

First, authors suggest “Qualitative results further revealed the high feasibility and acceptability” (L24), but it is difficult to understand why they could conclude “high” without Quantitative results. Authors should revise the manuscript, carefully.

Second, authors suggest “The thematic analysis of the qualitative data revealed three major themes: (1) high feasibility and acceptability, (2) perceived benefits, and (3) areas of improvement as reported in the session feedback questionnaires or focus groups. Overall, the qualitative analysis found that well-designed, accessible health education programs can effectively empower older adults with knowledge and strategies for maintaining brain health, potentially contributing to positive lifestyle changes and improving their quality of life.” (L291), “The high feasibility and acceptability of the workshops can be seen from these three key aspects: (1) appropriate timing and duration, (2) a comprehensive and interactive delivery structure, and (3) provision of scaffolding.” (L298), “These comments suggest that scheduling based on stakeholder feedback played a crucial role in making the program more accessible, which consequently increased its feasibility and acceptability among participants.” (L310), “This statement highlights the perceived effectiveness of the workshop structure, emphasizing its logical organization and alignment with individual learner’s cognitive processes. The participant’s use of terms like “immediate recall and response” suggests that the structure facilitated active learning and engagement.” (L321), “This interactive element likely contributed to the program’s effectiveness by enabling participants to clarify concepts, share experiences, and reinforce their learning.” (L328), “Considering the unique needs of the older Chinese American population, the workshops incorporated various scaffolding techniques. These included providing hardcopy materials, adapting the instructors’ speaking pace with repetition when necessary, and utilizing real-life examples to facilitate comprehension.” (L331), “The analysis revealed that participants strongly intended to modify their lifestyles following the workshops to maintain brain health. This included the willingness to make dietary changes, undergo regular health check-ups, exercise regularly, improve sleep quality, and maintain social activities.” (L349), “These accounts demonstrate how the workshops successfully raised participants’ awareness of brain health and motivated participants to adopt proactive and meaningful lifestyle changes aimed at reducing dementia risk, highlighting the program's positive impact on preventative health behaviors.” (L366), “The analysis yielded that, upon completion of the workshops, most participants reported an improvement in their health literacy, such as a better understanding of nutritional concepts, increased scientific knowledge about specific dietary components (such as proteins and fatty acid types), and a greater appreciation for the importance of dietary diversity.” (L370), “This account illustrates how integrating mindfulness techniques into daily life can lead to tangible improvements, such as enhanced sleep quality, showcasing the practical value of the program.” (L395), “Even though our quantitative study did not show a statistically significant decrease in dementia worry, qualitative results indicated that the workshops also reduced participants’ anxiety about dementia, positively impacting their overall well-being. A key factor in this reduction was the improved ability to distinguish between normal age-related cognitive decline and dementia symptoms.” (L398), “This quote demonstrates how increased knowledge led to a more balanced view of cognitive aging, reducing fear and promoting a more proactive approach to brain health. The workshops have empowered participants with information that allowed them to contextualize their experiences and concerns within a deeper understanding of cognitive aging.” (L409), “Three major areas of improvement were suggested by participants: (1) incorporating bilingual study materials, (2) including more experience-sharing components, and (3) adding more time for “Q & A”.” (L415), “Those responses underscore participants’ enthusiasm for peer experience-sharing to build a support network and deepen their learning.” (L443), and “This suggestion highlights the value participants place on interactive learning and their desire for opportunities to engage more deeply with the material presented.” (L453), but these descriptions should be in the discussion section. Authors should revise the manuscript, carefully.

Finally, authors described some of sentences without citation or justification as follows; “Chinese Americans, especially those 50 years or older and with limited English proficiency, face unique barriers to accessing health promotion programs. Specifically, cultural and linguistic challenges hinder their participation and engagement in these educational programs.” (L56), “Unfortunately, health information is often unavailable in a Chinese language or dialect.” (L61), and “While some qualitative data alluded to the program’s impact on reducing dementia worry, the quantitative results did not show significant changes.” (L509), but it is difficult for readers to judge them without references as evidence for each description. Authors should add references for these descriptions.

Author Response

Comment 1: First, authors suggest “Qualitative results further revealed the high feasibility and acceptability” (L24), but it is difficult to understand why they could conclude “high” without Quantitative results. Authors should revise the manuscript, carefully.

Response 1: Thank you for the comment. We accepted the suggestion and replaced the term "high" with "promising" to more accurately reflect the qualitative nature of our findings and avoid implying quantitative measurement.

Comment 2: Second, authors suggest “The thematic analysis of the qualitative data revealed three major themes: (1) high feasibility and acceptability, (2) perceived benefits, and (3) areas of improvement as reported in the session feedback questionnaires or focus groups. Overall, the qualitative analysis found that well-designed, accessible health education programs can effectively empower older adults with knowledge and strategies for maintaining brain health, potentially contributing to positive lifestyle changes and improving their quality of life.” (L291), “The high feasibility and acceptability of the workshops can be seen from these three key aspects: (1) appropriate timing and duration, (2) a comprehensive and interactive delivery structure, and (3) provision of scaffolding.” (L298), “These comments suggest that scheduling based on stakeholder feedback played a crucial role in making the program more accessible, which consequently increased its feasibility and acceptability among participants.” (L310), “This statement highlights the perceived effectiveness of the workshop structure, emphasizing its logical organization and alignment with individual learner’s cognitive processes. The participant’s use of terms like “immediate recall and response” suggests that the structure facilitated active learning and engagement.” (L321), “This interactive element likely contributed to the program’s effectiveness by enabling participants to clarify concepts, share experiences, and reinforce their learning.” (L328), “Considering the unique needs of the older Chinese American population, the workshops incorporated various scaffolding techniques. These included providing hardcopy materials, adapting the instructors’ speaking pace with repetition when necessary, and utilizing real-life examples to facilitate comprehension.” (L331), “The analysis revealed that participants strongly intended to modify their lifestyles following the workshops to maintain brain health. This included the willingness to make dietary changes, undergo regular health check-ups, exercise regularly, improve sleep quality, and maintain social activities.” (L349), “These accounts demonstrate how the workshops successfully raised participants’ awareness of brain health and motivated participants to adopt proactive and meaningful lifestyle changes aimed at reducing dementia risk, highlighting the program's positive impact on preventative health behaviors.” (L366), “The analysis yielded that, upon completion of the workshops, most participants reported an improvement in their health literacy, such as a better understanding of nutritional concepts, increased scientific knowledge about specific dietary components (such as proteins and fatty acid types), and a greater appreciation for the importance of dietary diversity.” (L370), “This account illustrates how integrating mindfulness techniques into daily life can lead to tangible improvements, such as enhanced sleep quality, showcasing the practical value of the program.” (L395), “Even though our quantitative study did not show a statistically significant decrease in dementia worry, qualitative results indicated that the workshops also reduced participants’ anxiety about dementia, positively impacting their overall well-being. A key factor in this reduction was the improved ability to distinguish between normal age-related cognitive decline and dementia symptoms.” (L398), “This quote demonstrates how increased knowledge led to a more balanced view of cognitive aging, reducing fear and promoting a more proactive approach to brain health. The workshops have empowered participants with information that allowed them to contextualize their experiences and concerns within a deeper understanding of cognitive aging.” (L409), “Three major areas of improvement were suggested by participants: (1) incorporating bilingual study materials, (2) including more experience-sharing components, and (3) adding more time for “Q & A”.” (L415), “Those responses underscore participants’ enthusiasm for peer experience-sharing to build a support network and deepen their learning.” (L443), and “This suggestion highlights the value participants place on interactive learning and their desire for opportunities to engage more deeply with the material presented.” (L453), but these descriptions should be in the discussion section. Authors should revise the manuscript, carefully.

Response 2:  We appreciate the suggestion regarding the placement of thematic analysis in the manuscript. However, thematic analysis involves more than organizing codes and reporting themes; it also includes researcher interpretation within the context. This approach is aligned with the framework proposed by Braun and Clarke (2006), which is widely recognized and cited in the field, including our manuscript. Based on this rationale, we have opted to retain these interpretations in the Qualitative Findings section.

Comment 3: Finally, authors described some of sentences without citation or justification as follows; “Chinese Americans, especially those 50 years or older and with limited English proficiency, face unique barriers to accessing health promotion programs. Specifically, cultural and linguistic challenges hinder their participation and engagement in these educational programs.” (L56), “Unfortunately, health information is often unavailable in a Chinese language or dialect.” (L61), and “While some qualitative data alluded to the program’s impact on reducing dementia worry, the quantitative results did not show significant changes.” (L509), but it is difficult for readers to judge them without references as evidence for each description. Authors should add references for these descriptions.

Response 3: Thank you for your comment. We accepted the suggestion and added references to substantiate the mentioned statements in our revised manuscript. We would like to clarify that the statement on qualitative and quantitative data regarding dementia worry reflects our findings and does not require external references. To make this clearer, we have rephrased it to "While our qualitative data alluded to the program’s impact on reducing dementia worry, our quantitative results did not show significant changes."

Round 2

Reviewer 2 Report

Comments and Suggestions for Authors

Thank you for your revised manuscript. I have no further comments. the table format should follow the journal requirement.

Author Response

Thank you for your feedback and for reviewing our revised manuscript. We have carefully reviewed the tables and confirm that they align with the journal’s requirements: https://www.mdpi.com/journal/geriatrics/instructions.

Reviewer 4 Report

Comments and Suggestions for Authors

Authors revised the manuscript, but this article has not fully answered some of the questions due to insufficient description.

In fact, as mentioned in the previous review, these are these descriptions which should be in the discussion section (i.e., “thematic analysis” part). Authors suggest “thematic analysis involves more than organizing codes and reporting themes; it also includes researcher interpretation within the context.”, but that is no reason to mix up results and interpretations. Authors should revise the manuscript, carefully.

Author Response

Comment:  Authors revised the manuscript, but this article has not fully answered some of the questions due to insufficient description. In fact, as mentioned in the previous review, these are these descriptions which should be in the discussion section (i.e., “thematic analysis” part). Authors suggest “thematic analysis involves more than organizing codes and reporting themes; it also includes researcher interpretation within the context.”, but that is no reason to mix up results and interpretations. Authors should revise the manuscript, carefully.

Response: 

Thank you for your continued feedback. We would like to clarify that our presentation of themes and subthemes, along with their interpretations, follows the well-established qualitative research reporting standards outlined by Braun and Clarke (2006), O’Brien et al. (2014), and the American Psychological Association (APA, 2019). According to these guidelines, thematic analysis involves not only identifying themes but also interpreting them within the Results section. The APA’s Journal Article Reporting Standards (Levitt et al., 2018) explicitly state that qualitative findings should include the meaning and understandings derived from the analysis, while the Discussion section should focus on the contribution and application of these findings.

In our previous response, we cited Braun and Clarke (2006), a seminal work with over 200,000 citations, to justify our approach. However, it appears this point was not fully considered. Moving our interpretation to the Discussion section, as suggested, would not align with qualitative research best practices and could compromise the manuscript’s scientific rigor. This structure is widely adopted by major journals publishing qualitative research.

Additionally, we note that this concern was not raised by other reviewers, further suggesting that our approach aligns with commonly accepted qualitative research standards. We have carefully reviewed our manuscript to ensure it adheres to these reporting guidelines. If the journal has specific formatting requirements that differ from these established conventions, we would appreciate clarification. Otherwise, we respectfully maintain our current structure to uphold the integrity of our qualitative findings.

References:

  • Braun, V., & Clarke, V. (2006). Using thematic analysis in psychology. Qualitative Research in Psychology, 3(2), 77-101.
  • Levitt, H. M., Bamberg, M., Creswell, J. W., Frost, D. M., Josselson, R., & Suárez-Orozco, C. (2018). Journal article reporting standards for qualitative research in psychology: The APA publications and communications board task force report. American Psychologist, 73(1), 26-46.
  • O’Brien, B. C., Harris, I. B., Beckman, T. J., Reed, D. A., & Cook, D. A. (2014). Standards for reporting qualitative research: a synthesis of recommendations. Academic Medicine, 89(9), 1245-1251.